# Highly Active Nanocrystalline ZnO and Its Photo-Oxidative Properties towards Acetone Vapor

**DOI:** 10.3390/mi14050912

**Published:** 2023-04-23

**Authors:** Artem Chizhov, Pavel Kutukov, Alexander Gulin, Artyom Astafiev, Marina Rumyantseva

**Affiliations:** 1Chemistry Department, Moscow State University, Moscow 119991, Russia; pavel.kutukov@chemistry.msu.ru (P.K.); roum@inorg.chem.msu.ru (M.R.); 2N.N. Semenov Federal Research Center for Chemical Physics of Russian Academy of Sciences, Moscow 119991, Russiaastafiev.artyom@gmail.com (A.A.)

**Keywords:** zinc oxide, hydrozincite, photocatalysis, acetone

## Abstract

Zinc oxide is one of the well-known photocatalysts, the potential applications of which are of great importance in photoactivated gas sensing, water and air purification, photocatalytic synthesis, among others. However, the photocatalytic performance of ZnO strongly depends on its morphology, composition of impurities, defect structure, and other parameters. In this paper, we present a route for the synthesis of highly active nanocrystalline ZnO using commercial ZnO micropowder and ammonium bicarbonate as starting precursors in aqueous solutions under mild conditions. As an intermediate product, hydrozincite is formed with a unique morphology of nanoplates with a thickness of about 14–15 nm, the thermal decomposition of which leads to the formation of uniform ZnO nanocrystals with an average size of 10–16 nm. The synthesized highly active ZnO powder has a mesoporous structure with a BET surface area of 79.5 ± 4.0 $m^{2}/g$, an average pore size of 20 ± 2 nm, and a cumulative pore volume of 0.507 ± 0.051 ${\mathrm{cm}}^{3}/g$. The defect-related PL of the synthesized ZnO is represented by a broad band with a maximum at 575 nm. The crystal structure, Raman spectra, morphology, atomic charge state, and optical and photoluminescence properties of the synthesized compounds are also discussed. The photo-oxidation of acetone vapor over ZnO is studied by in situ mass spectrometry at room temperature and UV irradiation ($\lambda_{\max}$ = 365 nm). The main products of the acetone photo-oxidation reaction, water and carbon dioxide, are detected by mass spectrometry, and the kinetics of their release under irradiation are studied. The effect of morphology and microstructure on the photo-oxidative activity of ZnO samples is demonstrated.

## 1. Introduction

Zinc oxide with a wurtzite structure is a wide-gap ($E_{g}∼3.3$ eV) *n*-type semiconductor [1] that also exhibits outstanding photocatalytic properties, owing to which it can compete with TiO_2_. The photocatalytic activity of ZnO has received interest for technologies such as photoactivated gas sensing [2,3], water splitting [4], photocatalytic synthesis [5,6], and air and water purification [7,8,9]. Moreover, defect engineering methods have been widely developed for ZnO that increase its reactivity by introducing additional surface oxygen vacancies during synthesis or physical treatment [10,11]. Compared with other metal oxides used as materials for gas sensors (SnO_2_, In_2_O_3_), ZnO has photocatalytic properties, which, for example, were demonstrated using photoactivated isotopic oxygen exchange [12]. Due to its nontoxicity, environmental stability, ecofriendliness, abundance, and low cost, ZnO can be considered as a material for wide practical applications.

The photocatalytic properties of ZnO depend on many aspects, such as particle size and morphology, the presence of impurities of other atoms, defects in the surface structure, among others. Various methods can be used for the synthesis of ZnO nanostructures, including the sol-gel method [13,14,15], hydrothermal methods [16,17], the thermal decomposition of precursors [18,19], and others [20].

One of the most preferred precursors for the synthesis of nanocrystalline ZnO by thermal decomposition is hydrozincite (HZ), which has a formula of Zn_5_(CO_3_)_2_(OH)_6_ (or 2ZnCO_3_·3Zn(OH)_3_). There are several reasons for this preference: (i) the decomposition of HZ proceeds at a relatively low temperature of 240–260 °C, which produces ZnO with a low crystallite size; (ii) the decomposition of HZ proceeds with the formation of a large volume of gases, which produces porous powders with a high specific surface area; (iii) the decomposition products CO_2_ and H_2_O do not contaminate the formed ZnO and are completely removed in the form of gases; (iv) HZ can efficiently accumulate heavy metal ions during coprecipitation [21], which enables the control of the introduction of impurities of various elements during the synthesis of ZnO.

Conventional laboratory methods used for the synthesis of HZ are based on the controlled hydrolysis of zinc salts in the presence of urea [22,23,24,25,26,27,28] or hexamethylenetetramine [29], or precipitation with alkali metal carbonates [30,31,32]. These methods require long heating and solvo- or hydrothermal conditions, which may be unacceptable for large-scale synthesis. More methods suitable for the synthesis of HZ have been proposed that are based on the reaction of metallic Zn or ZnO with CO_2_ and NH_3_ in an aqueous medium [33,34,35,36]. However, these methods required mechanical activation of precursor powders or gas equipment, which complicates the synthetic setup.

Acetone (2-propanone) is the simplest ketone that is, under normal conditions, a highly volatile and flammable liquid. Acetone vapors are dangerous to human health and are explosive when mixed with air in certain proportions. Typical symptoms of human exposure to acetone vapor are depression of the central nervous system and irritation of the mucous membranes of the eyes, nose, and throat, which appear at concentrations starting from 237 ppm (574 $\mathrm{mg}/m^{3}$) when exposed for several hours [37,38]. At the same time, the range of concentration of acetone vapor in the air to produce an explosion is 2.5–12.8 vol.%. The presence of acetone vapor in the air-exhaled breath at a level of several parts per million is a marker of various diseases, such as diabetes, so the detection of acetone at a low level is of great importance for noninvasive and quick medical diagnosis.

Resistive-type gas sensors are promising for detecting acetone in air, because they have high sensitivity, a simple design, and a wide measuring range [39,40]. Some of the advanced gas-sensitive materials used for acetone detection are ZnO and materials based on it [41,42,43,44,45]. In the last decade, an approach using UV or visible irradiation instead of thermal activation to promote the gas-sensitive properties of metal oxides has been developed [46,47]. The study of the mechanism of the photochemical reactions of molecules on the surface of semiconductors is of great importance for the targeted development of materials for photoactivated gas detection and photocatalysis [48,49].

ZnO and ZnO-based materials are promising materials that actively promote various photochemical reactions on their surface under UV or visible irradiation, which have been actively studied recently [50,51,52,53,54,55,56]. However, a detailed and systematic study of the some mechanisms, in particular, the photochemical oxidation of acetone, on the ZnO surface has not yet been carried out. An analysis of the literature shows that studies of the gas-phase photo-oxidation of acetone has mainly focused on TiO_2_ and TiO_2_-based materials [57,58,59], while information on the photo-oxidation of acetone on the ZnO surface is limited. For example, Ref. [60] presents the results of a study of acetone photo-oxidation over ZnO under mercury lamp light irradiation. In the cited work, two samples of ZnO with an average crystallite size of 20.9–41.1 nm (BET surface area 49.8 $m^{2}/g$) and 27.5–156.0 nm (BET surface area 13.2 $m^{2}/g$) were synthesized. The study of the kinetics of the photo-oxidation of acetone by gas chromatography was carried out according to the rate of release of the main oxidation product (CO_2_). The authors concluded that the rate of acetone photo-oxidation increases with increasing temperature from 40 to 240 °C; in addition, a sample with a larger grain size exhibited higher photo-oxidative activity.

In this paper, we propose a strategy for the simple wet synthesis of nanocrystalline and mesoporous ZnO powders using commercial ZnO micropowder and NH_4_HCO_3_ aqueous solution as starting precursors. The key feature of this method is the formation of HZ nanoparticles with the morphology of nanoplates, the thermal decomposition of which leads to the synthesis of nanocrystalline ZnO with a mesoporous structure. The crystal structure, morphology, charge state of atoms, optical properties, and other characteristics of the synthesized samples were studied. The photo-oxidation of acetone vapors was studied over ZnO under UV irradiation through in situ mass spectrometry at room temperature. The high efficiency of the synthesized ZnO in the photo-oxidation of acetone vapor was observed, which shows the promise of using this material in the fields of gas sensors, air purification, photocatalysis, among other applications.

## 2. Materials and Methods

### 2.1. Synthesis of Nanocrystalline ZnO

Nanocrystalline ZnO was synthesized in several steps:(1)A total of 50 g of commercial ZnO (ACS Reagent, ≥99.0%) was placed in a 3 L beaker, and 1.75 L of deionized water was added to it. The resulting suspension was stirred at 400 rpm using an overhead stirrer at a temperature of 22–25 °C for 30 min to increase the hydration of the surface of the ZnO particles and enhance their reactivity.(2)Then, 500 g of NH_4_HCO_3_ (BioUltra, ≥99.5%) was poured into a ZnO suspension, which was accompanied by foaming and cooling of the reaction mixture by 10–15 °C. Stirring was continued for 90 min, after which the resulting precipitate was sedimented and washed three times with deionized water by decantation. The precipitate obtained was marked as “ZnO-Prec(1)”.(3)The washed precipitate ZnO-Prec(1) was again suspended in 1 L of deionized water; the resulting suspension was heated up to 90 °C until no more gas evolved. After cooling and sedimenting, the precipitate was washed three times with deionized water, separated on a Buchner funnel, and washed with additional water and ethanol. The precipitate obtained was marked as “ZnO-Prec(2)”.(4)The ZnO-Prec(2) precipitate (and samples of ZnO-Prec(1) for further characterization) was dried at a temperature of 70 °C in air overnight and then annealed at a temperature of 300 °C in air for 24 h. The yield of nanocrystalline ZnO was approximately 86%.

### 2.2. Characterization of Samples

The phase composition and crystal structure of the synthesized precipitates and ZnO powders were studied by powder X-ray diffraction (XRD) with a Rigaku Max-2500 diffractometer (Rigaku Corporation, Tokyo, Japan) with a rotating anode (CuKα radiation; λ = 1.54059 Å). The refinement of the cell parameters was carried out with the nonlinear least squares method with regression diagnostics [61]. The average size of the coherent scattering regions $d_{s}$ was calculated using the Sherrer equation:(1)$$d_{s}=\frac{k\lambda }{\sqrt{\beta_{exp}^{2}-\beta_{app}^{2}}\cos\theta }$$
where λ is the wavelength of the X-ray radiation, nm; $\beta_{exp}$ is the observed peak width at half height; $\beta_{app}$ is the instrumental broadening, rad; θ is a diffraction angle; *k* is a coefficient that is equal to 0.9.

The Raman spectra were registered with a SENTERRA Raman microscope–spectrometer (Bruker, Billerica, MA, USA) using a 50 × 0.75 NA microscope objective lens and laser excitation at 785 nm.

The thermal analysis and determination of decomposition products were carried out on a Netzsch STA 409 PC Luxx synchronous thermal analyzer in combination with a Netzsch QMS 403 C Aëolos quadrupole mass spectrometer. Samples were heated in an air flow (30 mL/min) in the temperature range of 30–1000 °C, which was increased at a rate of 5 °C/min.

The morphology of the ZnO and HZ powders was studied with a scanning electron microscope (SEM) Prisma E (Thermo Scientific, Prague, Czech Republic). The samples were preliminary coated with a 10 nm layer of gold with a Q150R ES plus sputter coater (Quorum Technologies, Laughton, Great Britain). The study using high-resolution transmission electron microscopy (HR-TEM) and selected area electron diffraction (SEAD) was carried out with a microscope JEOL JEM-2100 (JEOL, Akishima, Tokyo, Japan). The specific surface area and average pore size were determined using an ASAP 2010 adsorption analyzer (Micromeritics, Norcross, GA, USA) with low-temperature nitrogen adsorption (BET model).

X-ray photoelectron spectroscopy (XPS) measurements were obtained using a OMICRON ESCA+ spectrometer (Scienta Omicron, Uppsala, Sweden) with an Al Kα X-ray source (E=1486.6 eV, 252 W). To compensate for the local charging of the analyzed surface, a CN-10 charge neutralizer was used with an emission current of 6 μA and a beam energy of 1 eV. The analyzer pass energy was 20 eV. The spectrometer was calibrated against the $\mathrm{Au}4f_{7/2}$ line at 84.1 eV. The main state of the C1s core level was used as a reference with a binding energy (BE) of 284.8 eV for the obtained spectra. FitXPS software was used for the deconvolution of the X-ray photoelectron spectra. The peaks were fitted with Gaussian–Lorentzian functions with a variable ratio (G/L), and the background was determined with the Shirley function.

The diffuse reflectance spectra were recorded in the wavelength range of 300–800 nm on a Perkin–Elmer Lambda 35 spectrophotometer (PerkinElmer, Inc., Waltham, MA, USA). The absorption spectra were recalculated using the Kubelka–Munk function (F) according to the following equation:(2)$$F=\frac{{\left(1-R\right)}^{2}}{2R}$$
where *R* is the diffuse reflectance coefficient.

Photoluminescence was excited with frequency-doubled pulses of a femtosecond titanium–sapphire oscillator (Tsunami, Spectra-Physics, Moscow, Russian Federation), with a central wavelength of 360 nm, repetition rate of 80 MHz, duration of 100 fs, and pulse energy of 10 pJ. The luminescence emission spectra were recorded with an EM-CCD camera PI MAX 2 (Princeton Instruments, Trenton, NJ, USA), installed at the monochromator output. The PL spectra of the samples were recorded at room temperature.

Fourier transform infrared spectroscopy (FTIR) studies were performed using a Perkin Elmer Frontier spectrometer (Perkin Elmer Inc., Beaconsfield, UK). The spectra were recorded in transmission mode in the wavenumber range of 4000–400 ${\mathrm{cm}}^{-1}$ with a resolution of 4 ${\mathrm{cm}}^{-1}$. To record the spectra, 0.5 mg of powdered samples was ground with 50 mg of KBr and pressed into tablets that were 7 mm in diameter.

### 2.3. Gas-Phase Photo-Oxidation of Acetone

A schematic diagram of the experimental setup is shown in Figure 1. To study the photo-oxidation of acetone vapor, we used a flow-type cell, the design of which was described in our previous paper [62]. ZnO samples in the form of tablets that were 20 mm in diameter and 1 mm thick were placed in a cell and hermetically sealed with a quartz window, which allowed the samples to be irradiated with UV light. As a source of acetone vapor, a microflow source was used, which was thin-walled PTFE tubes filled with acetone, and placed under thermostatic conditions (60 °C) in a flow cell. A carrier gas (O_2_/He mixture; oxygen concentration 100 ppm) was passed through the source of the microflow at a rate of 15 mL/min, which yielded about 50 ppm of acetone (determined gravimetrically). Next, the resulting gas mixture, 100 ppm O_2_ + 50 ppm acetone + He, was turned to a cell with ZnO samples, the outlet of which was connected to the capillary of a mass spectrometer (MS7–200, equipped with an RGA–200 analyzer, Stanford Research Systems) to monitor the concentrations of the molecules in the gas phase. A set of UV LEDs ($\lambda_{\max}$ = 365 nm) was used for UV irradiation; the irradiated surface area of the ZnO samples was 12.5 ${\mathrm{cm}}^{2}$ for each experiment; the irradiance of he samples was 40–60 $\mathrm{mW}/{\mathrm{cm}}^{2}$ (measured with a Nova II radiometer equipped with a photodiode PD300-UV-193 head, Ophir, Jerusalem, Israel).

## 3. Results

### 3.1. Phase Composition

Figure 2 shows the powder diffraction patterns of the intermediate precipitates and ZnO synthesized at 300 °C. According to the obtained data, ZnO-prec(1) was a two-phase sample, consisting of the main phase of orthorhombic zinc ammine carbonate ZnNH_3_CO_3_, a space group $Pna2_{1}$ (01-077-5096 card, PDF4), and a small amount of ZnO with a wurtzite structure (card 36-1451, PDF2). The refined unit cell parameters of the synthesized ZnNH_3_CO_3_ were a=9.1488(11), b=7.6000(12), and c=5.4983(7). The ZnO-prec(2) sample was single-phase monoclinic hydrozincite Zn_5_(CO_3_)_2_(OH)_6_, with a space group C2/m (card 19-1458, PDF2), with refined cell parameters a=13.6127(19), b=6.3301(12), c=5.3788(5), and β=95.06(2). The annealing of the monoclinic HZ phase led to the formation of a single-phase sample of hexagonal ZnO, space group $P6_{3}mc$ (card 36-1451, PDF2). The refined unit cell parameters of synthesized ZnO were a=3.2490(14) and c=5.2073(20). A comparison of the XRD patterns of the synthesized nanocrystalline ZnO and micropowder ZnO precursor is shown in Appendix A.

Raman spectroscopy confirmed the phase composition of the synthesized compounds. The Raman spectrum of the synthesized ZnNH_3_CO_3_ (ZnO-Prec(1)) is shown in the Figure 3a. The most intense peaks were observed at 366, 432, 1097, and 1606 cm^−1^. Unfortunately, no data on the Raman spectra of zinc ammine carbonate were found in the literature. Figure 3b shows the Raman spectrum of the synthesized HZ (ZnO-Prec(2)). It shows the characteristic bands of HZ at 1061, 1368, and 1543 cm^−1^ and multiple bands at 734 and 707 cm^−1^ [63]. The characteristic band of smithsonite (ZnCO_3_) at 1092 cm^−1^ is not present in the spectrum. The Raman spectrum of the nanocrystalline ZnO (ZnO-300 sample) contained peaks at 104, 332, and 436 cm^−1^, which correspond to the $E_{2}^{\mathrm{low}}$, $E_{2}^{\mathrm{high}}-E_{2}^{\mathrm{low}}$, and $E_{2}^{\mathrm{high}}$ processes, respectively. These modes correspond to vibrations of the zinc (104 cm^−1^) and oxygen (436 cm^−1^) sublattices. The spectrum also contains broad peaks centered at 1070 and 1350 cm^−1^, which probably belong to second-order Raman modes [64].

Based on the phase analysis, the synthesis route could be described as follows: The reaction of ZnO with an aqueous solution of NH_4_HCO_3_ includes several stages; first, ammonium bicarbonate undergoes dissociation:(3)$$NH_{4}\mathrm{HC}O_{3}⇄N{H_{4}}^{+}+\mathrm{HC}{O_{3}}^{-}$$

Ammonium hydroxide and carbonic acid are a weak base and a weak acid, respectively, so the equilibrium shifts towards the formation of NH_3_ and CO_2_, which partially volatilize from the solution:(4)$$N{H_{4}}^{+}+{\mathrm{OH}}^{-}⇄{\mathrm{NH}}_{3}+H_{2}O$$
(5)$$\mathrm{HC}{O_{3}}^{-}+H_{3}O^{+}⇄CO_{2}+2H_{2}O$$

In a weakly basic solution, ZnO subdissolves in the presence of ammonia to form an amino complex:(6)$$\mathrm{ZnO}+4NH_{3}+H_{2}O⇄{\left[\mathrm{Zn}{\left({\mathrm{NH}}_{3}\right)}_{4}\right]}^{2+}+2{\mathrm{OH}}^{-}$$

The resulting complex ion can react with hydrocarbonate ions to form zinc ammine carbonate, which precipitates out from the solution:(7)$${\left[\mathrm{Zn}{\left({\mathrm{NH}}_{3}\right)}_{4}\right]}^{2+}+2\mathrm{HC}{O_{3}}^{-}\to \mathrm{Zn}NH_{3}CO_{3}↓\;+\;3NH_{3}+H_{2}O+CO_{2}$$

Although it was published earlier [65] that under close conditions, the reaction of ZnO with an aqueous NH_4_HCO_3_ solution leads to the one-stage formation of HZ, our studies showed that an intermediate formation of ZnNH_3_CO_3_ takes place. The traces of ZnO wurtzite phase, also revealed in the ZnO-Prec(1) precipitate by XRD (Figure 2a), may have been incompletely reacted initial ZnO micropowder; however, its presence was not confirmed in the Raman spectra (Figure 3a,b) or the XRD pattern of the ZnO-Prec(2) precipitate (Figure 2a). Another possible explanation is the partial decomposition of zinc ammine carbonate during XRD analysis or during storage to form the ZnO phase. Hot water treatment of zinc ammine carbonate leads to the quantitative formation of the HZ phase:(8)$$5\mathrm{ZnN}H_{3}CO_{3}+3H_{2}O\overset{90{}^{\circ}C}{\to }Zn_{5}{\left(CO_{3}\right)}_{2}{\left(\mathrm{OH}\right)}_{6}↓\;+\;3CO_{2}+5NH_{3}$$

The thermogravimetric analysis of synthesized HZ showed two main areas of weight loss (Figure 4). The first process (I) with a weight loss of about 5% was the desorption of chemisorbed H_2_O molecules from the surface of the HZ nanocrystals in the temperature range of 40–150 °C, which was accompanied by an endothermal effect. The desorption of water molecules in this temperature range was confirmed by the mass spectrum of water (m/z = 18 a.m.u.), which showed a broad peak centered at about 90 °C. The second process (II), which took place at 238 °C, corresponded to the decomposition of HZ according to:(9)$$Zn_{5}{\left(CO_{3}\right)}_{2}{\left(\mathrm{OH}\right)}_{6}\overset{300{}^{\circ}C}{\to }5\mathrm{ZnO}+2CO_{2}+3H_{2}O$$

The weight loss at HZ decomposition was 26.9% (excluding chemisorbed water), which is close to the theoretical value (25.9%). The calculated thermal effect of HZ decomposition is 150.92 kJ/mol. The mass spectra indicated that the decomposition of hydrozincite is a one-step process with the simultaneous release of CO_2_ and H_2_O molecules. An endothermic process (III) was observed on the DTA curve in the temperature range of 430–900 °C that was not accompanied by weight loss. The release of gaseous decomposition products apparently corresponded to the recrystallization and coarsening of the initially formed ZnO nanocrystals.

### 3.2. Morphology of Synthesized Materials

The synthesized HZ nanoparticles had a morphology of irregular-shaped nanoplates, as can be seen from the TEM images. HZ nanoplates with a lateral size of 100–200 nm formed bunch-like micrometer-sized aggregates (Figure 5a–c). The thickness of the nanoplates, determined from individual particles located perpendicular to the TEM image plane, was 14–15 nm. High-magnification HR-TEM images that could show interplanar distances on the faces of the HZ nanoparticles were not obtained due to the fast degradation of the nanoplates under the electron beam. The crystallographic orientation of the HZ nanoparticles was estimated by the diffraction of the electrons on a single HZ nanoplate lying parallel to the TEM image plane (Figure 5d). The resulting SAED pattern showed that the electron beam was incident on the nanoplate along the *b* axis; thus, the surface of the nanoplate was formed by planes in the [010] direction. From the SAED pattern, the β angle was estimated for the hydrozincite monoclinic unit cell, which was equal to 93.6° and slightly differed from the angle given on card 19-1458 from the PDF2 database (95.6°). The estimation of the thickness of the nanoplate from the broadening of the (020) XRD reflection according to Equation (Equation 1) provided an average value of 12 ± 1 nm, which generally agreed with the data from microscopy. At the same time, for the other crystallographic directions, [002] and [200], the calculated $d_{s}$ values are 10 ± 1 and 15 ± 1 nm, respectively. The narrowest reflection in the diffraction pattern of HZ nanoplates with a calculated $d_{s}$ = 23 ± 1 nm corresponding to the (021) planes, generally indicated an anisotropic domain structure of the HZ nanoplates.

The morphology of ZnO nanoparticles is demonstrated in Figure 6. The HR-TEM images show aggregates of roundish ZnO nanocrystals without clearly defined faces (Figure 6a). The calculated d-spaces from the SAED pattern confirmed the ZnO wurtzite structure (Figure 6b). The size distribution of the ZnO nanoparticles (measured over a larger dimension of crystallite) followed an approximately log-normal distribution with an average crystallite size of 16 ± 4 nm (Figure 6c). On the contrary, the measurement of the elliptical particles perpendicular to the elongation axis produced an average size of 10 ± 3 nm. Figure 6d shows a HR-TEM image of a typical elliptical ZnO nanocrystal, for which a d-space 0.26 nm was determined, which corresponds to the distance between the (0002) planes; thus, the elliptical ZnO nanocryctals were usually elongated in the *c* direction. An estimate of $d_{s}$ according to Equation (Equation 1) provided close values of 10–13 nm for all planes except for the (002) plane, for which $d_{s}$ was slightly larger and amounted to 16 ± 1 nm, which also confirmed an anisotropic shape of the ZnO nanocrystals and exactly agreed with the TEM results.

Studies using the low-temperature nitrogen adsorption method showed an adsorption isotherm of the IV type, which corresponded to the mesoporous structure of the synthesized ZnO (Figure 7a). The specific surface area of the ZnO determined by the BET model was 79.5 ± 4.0 $m^{2}/g$. The average pore size determined by desorption was 20 ± 2 nm; the cumulative pore volume was 0.507 ± 0.051 ${\mathrm{cm}}^{3}/g$.

### 3.3. Optical Properties

Figure 7b shows the absorption spectrum of the synthesized nanocrystalline ZnO recalculated using Equation (Equation 2). The absorption edge of the nanocrystalline ZnO powder is observed near 400 nm; in the visible region, there is also a slight absorption band in the range of 450–600 nm. The optical band gap of the nanocrystalline ZnO calculated using the Tauc plot for direct allowed transitions was 3.18 eV. The PL spectra of the nanocrystalline ZnO under 360 nm excitation is also demonstrated in Figure 7b, which shows a broad defective luminescence band covering almost the entire visible range and centered near 575 nm. The peak corresponding to the exciton transition in ZnO was not observed, because the PL spectrum was cut off with a filter at 400 nm. The observed PL peak could be resolved into at least two peaks with maxima at 521 and 593 nm, the intensity ratio of which was approximately 1:4.8. The photoluminescence of ZnO in the visible spectrum was due to point defects in its crystal structure and on its surface [66]. Despite the fact that the interpretation of the PL spectra of ZnO is usually difficult due to the possibility of the existence of various defects in its structure, the green band (521 nm) can be attributed to the recombination of photoexcited holes in the valence band, with electrons captured by ionized oxygen vacancies. The orange band (593 nm) is more likely to be related to the process of electron recombination at the interstitial oxygen atoms [67,68,69].

### 3.4. Elemental Composition and Charge States of Atoms

The XP spectra of the synthesized ZnO in the C1s, N1s, Zn2p, and O1s regions are shown in Figure 8. The carbon in the sample showed the three ordinary states of adventitious carbon, of which the main one was used as a reference with a binding energy of 284.8 eV; the other two corresponded to the oxidized state of C in CO and CO_3_ groups. Nitrogen is represented by a N1s peak of very low intensity in a charge state, presumably related to the ammonium groups on the ZnO surface, because, for N atoms in a ZnO lattice, a lower BE is expected according to the literature [70]. The spectrum of zinc is represented by a Zn2p doublet (only a single $\mathrm{Zn}2p_{3/2}$ peak is shown in Figure 8c for clarity). The observed spectrum is not accurately described by a single peak, which indicated the presence of an additional peak with a low intensity and a slightly higher BE. The charge state of the main Zn2p(I) peak corresponds to lattice zinc, while Zn2p(II) is related to surface Zn atoms bonded to hydroxyl groups. The best fit with the experimental data for the O1s spectrum is achieved with three peaks, which correspond to the three charge states of oxygen. The O1s(I) state with the lowest BE and the highest intensity refers to lattice oxygen. The positions of the peaks on the BE scale, as well as the calculated atomic ratio of the elements in the sample on the basis of the XPS analysis, are presented in Table 1. As can be seen, there was a slight excess of the surface oxygen content compared with zinc (the Zn:O ratio was 1:1.12). An overstoichiometric oxygen content can be due to the presence of interstitial oxygen in the ZnO lattice, as well as additional oxygen from adsorbed water molecules. In this case, state O1s(III) may correspond to surface OH groups, and state O1s(II) corresponds to interstitial oxygen atoms. The assumption of the presence of interstitial oxygen in the ZnO lattice is in agreement with the PL spectroscopy data, which also indicated the O_i_-related luminescence of the synthesized nanocrystalline ZnO samples (Section 3.3). The survey XP spectrum of the synthesized nanocrystalline ZnO and XP spectra of the synthesized HZ nanoplates are given in the Appendix A.

### 3.5. Gas-Phase Photo-Oxidation Acetone over ZnO

The gas-phase UV-activated oxidation of the acetone vapor was studied with in situ mass spectrometry at room temperature over the synthesized highly active ZnO and the reference ZnO sample, obtained by the decomposition of a commercial HZ powder (Sigma-Aldrich, #96466) at 300 °C. The structure and morphology of this ZnO sample were characterized in our previous study [62]. Despite the fact that the reference ZnO sample originated from the same compound, it was distinguished by a low specific surface area (4 $m^{2}/g$) and larger crystallite sizes (15–220 nm), which allowed it to reveal the effect of the microstructure on the photo-oxidative properties of the ZnO samples. The irradiated surface area (12.5 ${\mathrm{cm}}^{2}$) of the synthesized and reference ZnO samples and their location in the cell were the same in both experiments, enabling a correct comparison of photo-oxidative properties. Immediately before the measurements, the ZnO tablets were annealed in air at 300 °C for 24 h.

Before UV irradiation, the oxygen–acetone–helium gas mixture was passed through the cell with ZnO samples for several hours in the dark until a steady state was established. Figure 9 shows the full-scan mass spectra in the range of 10–60 a.m.u. of the carrier gas passed over the synthesized ZnO sample in the dark and during UV irradiation. As shown, the UV irradiation significantly affected the concentration of some molecules, primarily acetone, carbon dioxide, oxygen, and water, in the carrier gas. A similar difference in the mass spectra in the dark and under UV irradiation was also observed when passing a carrier gas over the reference ZnO sample (Appendix A). At the same time, no intermediate products of acetone photo-oxidation were detected by mass spectrometry in either case.

Figure 10 shows the change in the concentrations of acetone, O_2_, CO_2_, and H_2_O molecules in the carrier gas during UV radiation. To monitor the concentration of the above molecules, the mass numbers corresponding to the maximum signal in their mass spectra were chosen as 43 a.m.u. for acetone (CH_3_CO fragment), 32 a.m.u. for O_2_, 44 a.m.u. for CO_2_, and 18 a.m.u. for H_2_O. In the initial period of irradiation, an increase in the concentration of acetone molecules in the carrier gas was observed, which indicated their photodesorption from the ZnO surface under UV irradiation (Figure 10a). Further, the concentration of acetone molecules in the carrier gas slowly decreased relative to the dark stationary value, which indicated their consumption from the gas phase during the oxidation process. Finally, when UV irradiation was turned off, a sharp peak in the acetone absorption from the gas phase was observed, which restored the initial (dark) concentration of adsorbed molecules on the ZnO surface. Although the curves of the changes in the acetone concentration in the carrier gas during UV irradiation for both the synthesized and reference ZnO samples had a similar shape, the photodesorption peak for the highly active ZnO was much weaker, while the decrease in the acetone concentration during the irradiation period, on the contrary, was greater than that for the reference ZnO sample. Thus, for the synthesized highly active ZnO, the photodesorption of acetone molecules was suppressed, which indicated the immediate involvement of preadsorbed acetone molecules in the photo-oxidation processes. On the other hand, most of the adsorbed acetone molecules on the reference ZnO sample were desorbed under UV irradiation without involving the photo-oxidation reaction, which demonstrated the lower reactivity of the reference ZnO sample in the process of acetone photo-oxidation than that of the synthesized nanocrystalline ZnO. Under prolonged irradiation, the conversion of acetone from the gas phase reached 29% for the synthesized highly active ZnO and 10% for the reference ZnO. It was also found that under oxygen-rich conditions (carrier gas composition: 21% O_2_ + 79% N_2_ + 50 ppm acetone), the conversion of acetone over the synthesized nanocrystalline ZnO reached 85% under similar irradiation conditions.

Figure 10b,c also show that the UV irradiation of the ZnO samples was accompanied by the consumption of oxygen from the carrier gas and the release of CO_2_, which confirmed the photo-oxidation of acetone on the surface of ZnO according to the following reaction:(10)$$CH_{3}(\mathrm{CO})CH_{3}+4O_{2}\overset{h\nu }{\to }3CO_{2}+3H_{2}O$$

At the same time, the overall rates of O_2_ consumption and CO_2_ release were significantly higher for the synthesized ZnO than for the reference ZnO sample, which also indicated its higher photocatalytic activity in the gas-phase oxidation of acetone. The decrease rates of the consumption of acetone molecules and the release of CO_2_ during the irradiation period showed a tendency to reach saturation under prolonged irradiation. Thus, the kinetics of the consumption of acetone and the release of CO_2_ were opposite to one another and were generally consistent with each other. However, the kinetics of oxygen and acetone consumption differed from each other because, in the initial period of irradiation, there was a large consumption of oxygen from the carrier gas, then the rate decreased and tended to reach a plateau. The rapid oxygen consumption may have been due to oxidation of preadsorbed acetone molecules on the surface of ZnO in the initial irradiation period. Presumably, the resulting CO_2_ molecules accumulated in the adsorbed form on the ZnO surface and gradually desorbed into the gas phase, which also affected the observed kinetics of CO_2_ release. Under UV irradiation, the concentration of the water molecules in the carrier gas also increased, but there was only a slight difference between the samples, which may have pointed to the predominant adsorption of water molecules formed during acetone photo-oxidation on the surface of ZnO (Figure 10d).

### 3.6. Discussion

In a previous work [12], we developed a model according to which, under UV irradiation, the surface of ZnO was enriched with oxygen vacancies through the photolysis of the surface layers. The generated oxygen vacancies acted as adsorption sites for oxygen molecules, which then underwent dissociation with the healing of the surface vacancies. Thus, the mechanism of acetone photo-oxidation on the surface of ZnO can be understood on the basis of the initial interaction of the adsorbed acetone molecules with photoexcited holes acting as oxidizers, with the participation of oxygen atoms [71]:(11)$$CH_{3}\mathrm{CO}CH_{3}+2h^{+}+7O\overset{h\nu }{\to }3CO_{2}+2H_{2}O+2H^{+}$$

To maintain electrical neutrality, protons are reduced by photoexcited electrons:(12)$$2H^{+}+2e^{-}\overset{h\nu }{\to }H_{2}$$

Although the formation of some intermediate or by-products of oxidation, such as formic acid, acetic acid, methylglyoxal, methanol, etc., could be expected as a result of the photo-oxidation of acetone, only the complete oxidation products were detected with mass spectrometry in the gas phase, indicating the strong oxidation potential of the photoexcited holes. The high efficiency of acetone photo-oxidation on the surface of the synthesized ZnO was apparently due to the fact that, with a grain size of 10–16 nm, the surface existed under highly nonequilibrium conditions, which facilitated the formation of oxygen vacancies on the surface under UV irradiation.

The studies performed on the gas-phase photo-oxidation of acetone demonstrated the importance of the morphology and microstructure for the reactivity of ZnO in photochemical reactions. The higher photo-oxidative activity of the synthesized nanocrystalline ZnO is a consequence of its morphology, which is formed as a result of the decomposition of the nanostructured precursor. Thus, the presented process is an example of how the morphology of precursors is “inherited” as a result of the formation of nanoparticles (similar examples can be found in Refs. [32,72,73]). The ultrafine thickness of the synthesized HZ nanoplates (14–15 nm) limits the size of ZnO grains formed during low-temperature decomposition (when recrystallization processes do not yet play a large role), and the small lateral size (up to 200 nm) controls the aggregation, leading to a highly porous structure. As a result, uniform ZnO nanoparticles with a narrow size distribution are formed, the combination of which to form aggregates gives a mesoporous structure with a pore size to the order of the size of nanocrystals. Although the synthesized ZnO had a high specific surface area, photochemical processes took place only on the irradiated surface, which was equal for both samples in the experiments; thus, the enhancement in the reactivity of the synthesized sample must have been due to some other parameter of the material. Because the synthesized ZnO nanocrystals did not have any obvious faceting, the higher reactivity could be generally attributed to their small grains size (10–16 nm) and the mesoporous microstructure of material. It should be noted that the XPS and PL spectroscopy studies demonstrated the same surface composition and defective structure of the synthesized and reference ZnO samples; therefore, the increase in the reactivity of the synthesized ZnO during acetone photo-oxidation could indeed be primarily explained by a decrease in the grain size.

The effect of the morphology of the HZ nanoparticles on the morphology of ZnO nanoparticles obtained by its thermal decomposition is be illustrated by the SEM images in Figure 11. The morphology of the synthesized HZ nanoparticles was also found in the SEM image as nanoplates (Figure 11a). As a result of the decomposition of HZ nanoplates, small ZnO nanoparticles formed, which stuck together, forming porous aggregates several microns in size (Figure 11b). On the other hand, the commercial HZ powder consisted of dense spherical micron-sized aggregates (Figure 11c), and their thermal decomposition also resulted in micrometer-sized aggregates of needle-shaped ZnO nanoparticles 15–30 nm thick and about 200 nm long (Figure 11d). A comparison of the XRD patterns and Raman and PL spectra of the synthesized and commercial samples of hydrozincite is shown in Appendix A

The studies carried out with IR spectroscopy also demonstrated that the surface compositions of the highly active synthesized ZnO and the reference ZnO were similar to each other. The FTIR spectra of both samples are shown in Figure 12. The main absorption band of the considered samples was located at 415 cm^−1^ (as well as a shoulder at 542 cm^−1^), which could be attributed to Zn–O stretching vibrations in the wurtzite crystal structure. The two weak absorption peaks at 1385 and 1625 cm^−1^ belonged to the symmetric and asymmetric vibrational modes of the carbonyl group, and the intensities of these peaks were very close in both samples. The broad peak with a maximum at 3435 cm^−1^ referred to the stretching vibrations of the OH groups on the surface of the ZnO grains, while the intensity of this band for the highly active ZnO was somewhat higher than that for the reference ZnO sample. This corresponded to a higher concentration of OH groups on the surface of the synthesized ZnO, which is consistent with its larger specific surface area compared with that of the reference ZnO sample. The FTIR spectra of some of the other compounds discussed in the article can be found in the Appendix A.

Thus, although hydrozincite is usually the preferred precursor for the synthesis of ZnO nanoparticles, its morphology is of great importance for obtaining ultra-dispersed and highly active ZnO. In this paper, we presented a method for the synthesis of highly active nanocrystalline mesoporous ZnO, which is based on common reagents and can be scaled up to obtain large amounts of the product. A particular advantage of the described synthesis method is that the high-purity nanocrystalline ZnO is not substantially contaminated by light element impurities, such as carbon and nitrogen, which was confirmed by XPS. For example, it is known from the literature that light elements easily dope ZnO if the precursor is annealed in the presence of an excess of C-, N-, and S-containing substances [74,75,76]. In this study, HZ nanoplates formed a single-phase, heavy, coarse-grained precipitate that was easily separated from a solution, which simplified its washing and the removal of traces of ammonia and NH_4_^+^ ions. Due to the lamellar shape of HZ nanoparticles, there are no internal cavities in which the supernatant liquid can enter, so it is possible to almost completely remove the NH_4_^+^ ions by washing the precipitate. The elimination of unintended doping pathways allowed us to obtain more accurate information about the physical and chemical properties of the synthesized samples, serving as the starting point for the controlled doping of synthesized materials.

## 4. Conclusions

In this paper, we showed that the heterophase reaction of ZnO with an excess of NH_4_HCO_3_ in an aqueous medium leads, first, to the formation of zinc ammine carbonate crystals. The hydrolysis of zinc amino carbonate in hot water, in turn, leads to the quantitative formation of hydrozincite with the morphology of ultrathin nanoplates 14–15 nm thick and a lateral size of about 200 nm. The decomposition of hydrozincite nanoplates in air at a temperature of 300 °C leads to the formation of nanocrystalline ZnO with a wurtzite structure and lattice parameters of a=3.2490(14) and c=5.2073(20). ZnO forms mesoporous structure consisted of rounded grains with an average size of 10–16 nm, a specific surface area of ZnO powder of 79.5 ± 4.0 $m^{2}/g$, and an average pore diameter of 20 ± 2 nm. The PL spectrum of ZnO in the visible range is represented by a wide band with a maximum located at 575 nm. The photo-oxidative properties of the synthesized ZnO were studied via in situ mass spectrometry at room temperature and UV irradiation, and the main oxidation products (CO_2_ and H_2_O molecules) were detected. The consumption of oxygen and acetone vapor from the carrier gas was registered. The synthesized highly active ZnO demonstrated about three times higher photo-oxidative activity than a reference ZnO sample with a low specific surface area and a larger crystallite size. The synthesized highly active ZnO can be considered a promising material for various photochemical applications, in particular, for photoactivated gas sensing.

## Figures and Tables

**Figure 1 micromachines-14-00912-f001:**
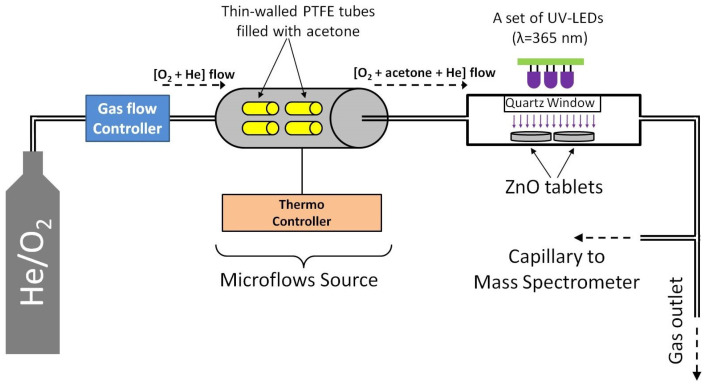
Schematic diagram of the experimental setup.

**Figure 2 micromachines-14-00912-f002:**
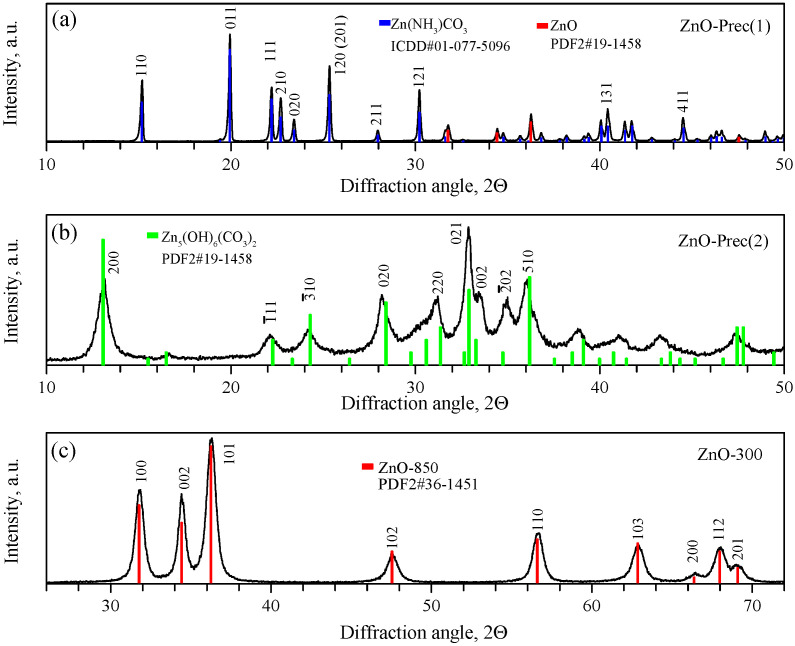
XRD patterns of ZnO-Prec(1) (**a**), ZnO-Prec(2) (**b**), and ZnO-300 (**c**) powders.

**Figure 3 micromachines-14-00912-f003:**
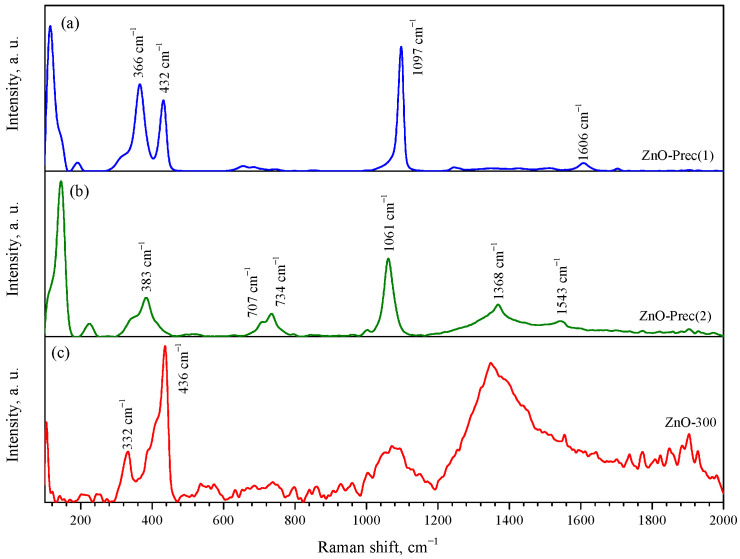
Raman spectra of ZnO-Prec(1) (**a**), ZnO-Prec(2) (**b**), and ZnO-300 (**c**) powders.

**Figure 4 micromachines-14-00912-f004:**
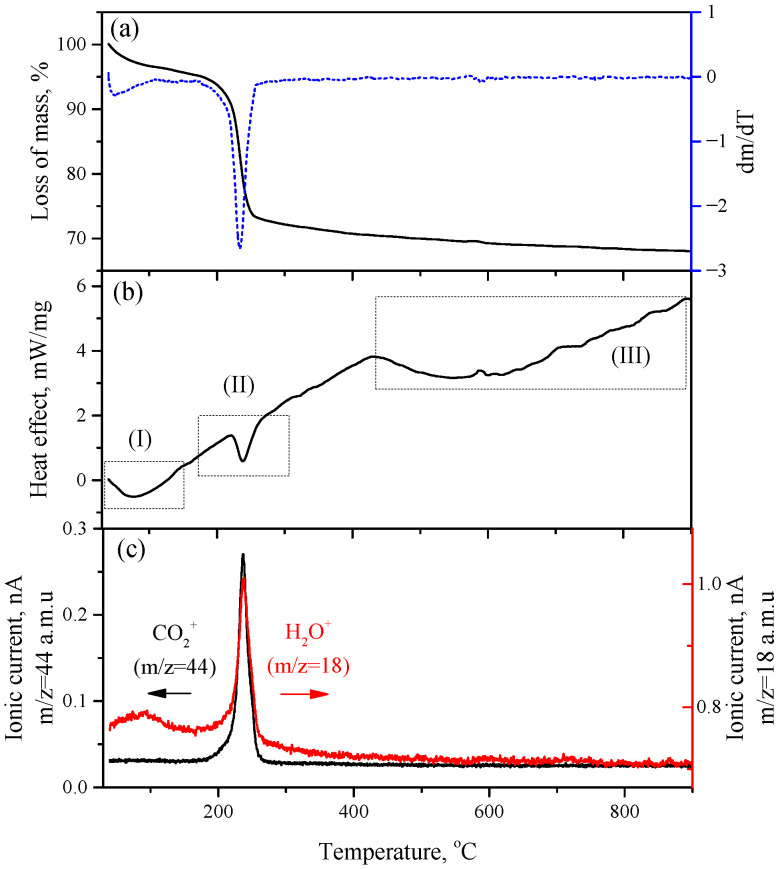
(**a**) Mass loss curve of synthesized HZ and its first derivative; (**b**) DTA curve of synthesized HZ; (**c**) mass spectrometric detection of decomposition products CO_2_ (m/z = 44 a.m.u) and H_2_O (m/z = 18 a.m.u).

**Figure 5 micromachines-14-00912-f005:**
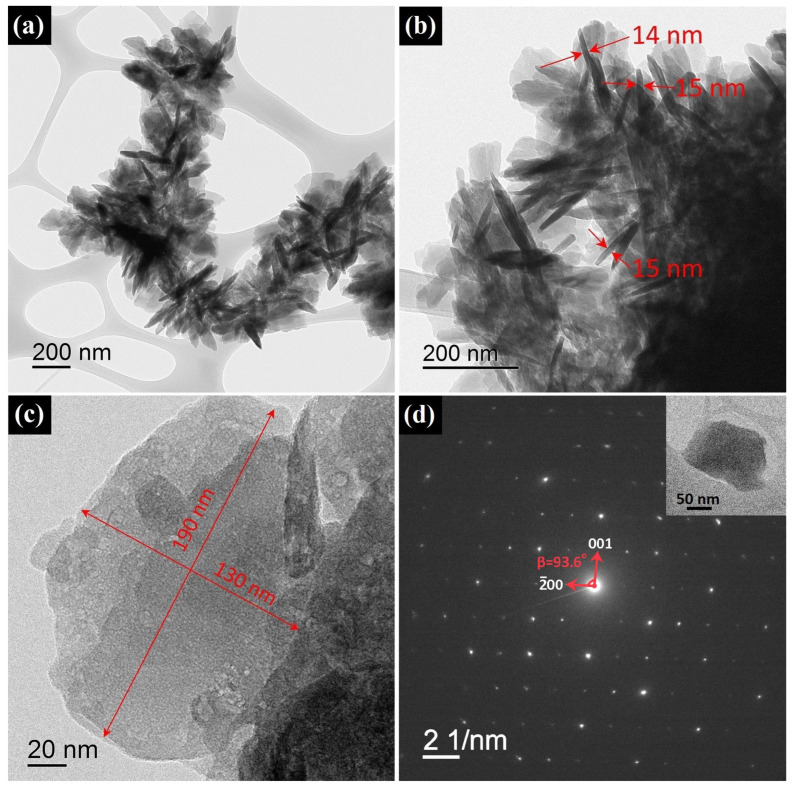
HR-TEM images of synthesized HZ nanoplates at various magnifications (**a**–**c**) and SEAD pattern from area shown in the inset (**d**).

**Figure 6 micromachines-14-00912-f006:**
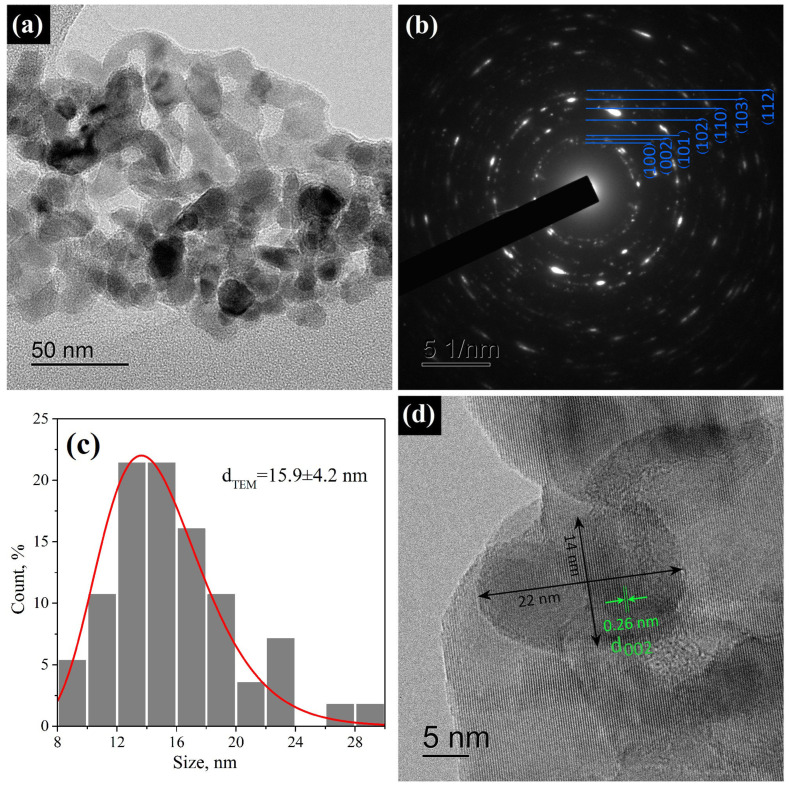
(**a**) HR-TEM image of aggregated ZnO nanoparticles; (**b**) SAED pattern of the area in (**a**); (**c**) size distribution of ZnO nanoparticls measured over a larger dimension of crystallite; (**d**) HR-TEM image of typical elliptical ZnO nanoparticle with marked (0002) d space on the face.

**Figure 7 micromachines-14-00912-f007:**
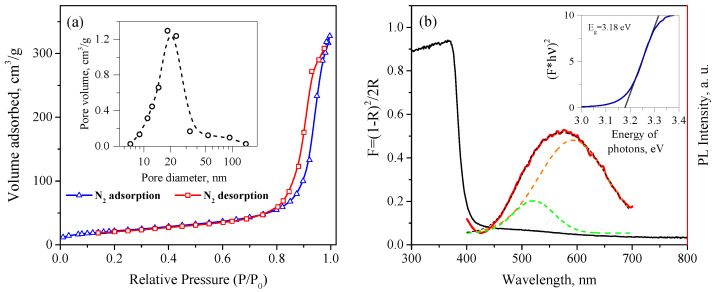
(**a**) N_2_ adsorption –desorption isotherms and pore size distribution (inset) of synthesized nanocrystalline ZnO powder; (**b**) Kubelka–Munk transform of diffuse reflectance spectrum of nanocrystalline ZnO (solid black line) and PL spectra at 360 nm excitation (red line); the inset shows the determination of the band gap of ZnO using the Tauc plot for direct transitions.

**Figure 8 micromachines-14-00912-f008:**
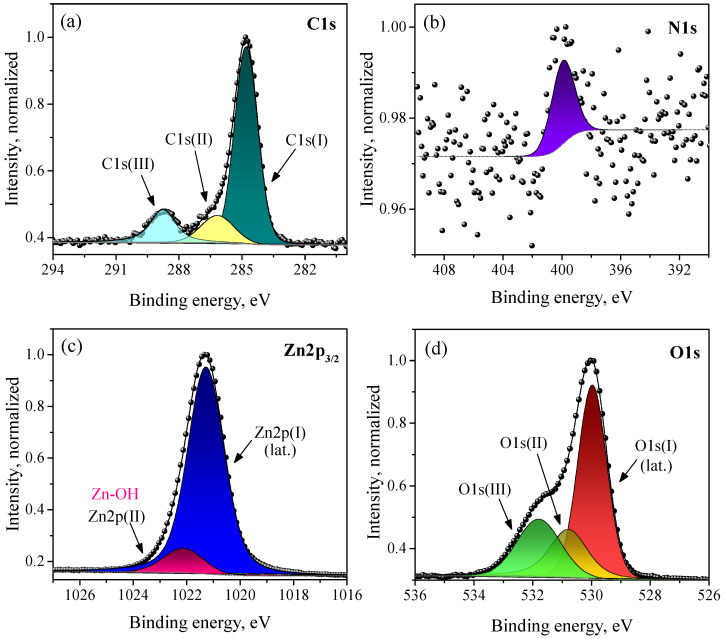
XP spectra of synthesized nanocrystalline ZnO on the C1s (**a**), N1s (**b**), $\mathrm{Zn}2p_{3/2}$ (**c**), and O1s (**d**) regions.

**Figure 9 micromachines-14-00912-f009:**
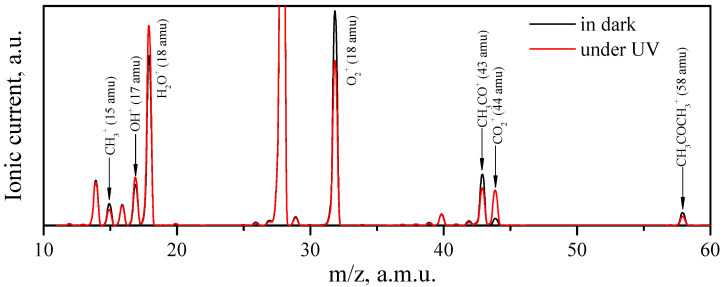
Full-scan mass spectra of the carrier gas exiting of the cell with synthesized ZnO in dark (black line) and under UV irradiation (red line) in the range of 10–60 a.m.u. Peak corresponding to the m/z = 28 a.m.u. (N_2_^+^ ions) was omitted due its high intensity.

**Figure 10 micromachines-14-00912-f010:**
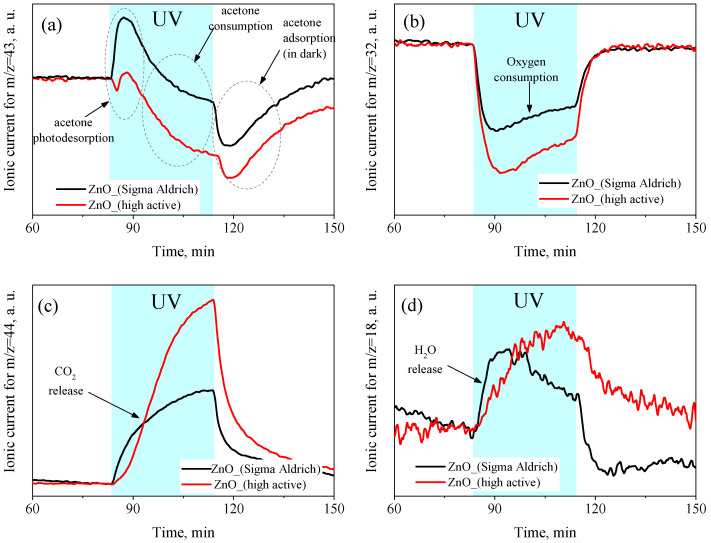
The concentration of CH_3_COCH_3_ (**a**), O_2_ (**b**), CO_2_ (**c**), and H_2_O (**d**) in the gas phase at the outlet of the cell during the photo-oxidation of acetone over the ZnO samples at room temperature, and UV irradiation determined by in situ mass spectrometry.

**Figure 11 micromachines-14-00912-f011:**
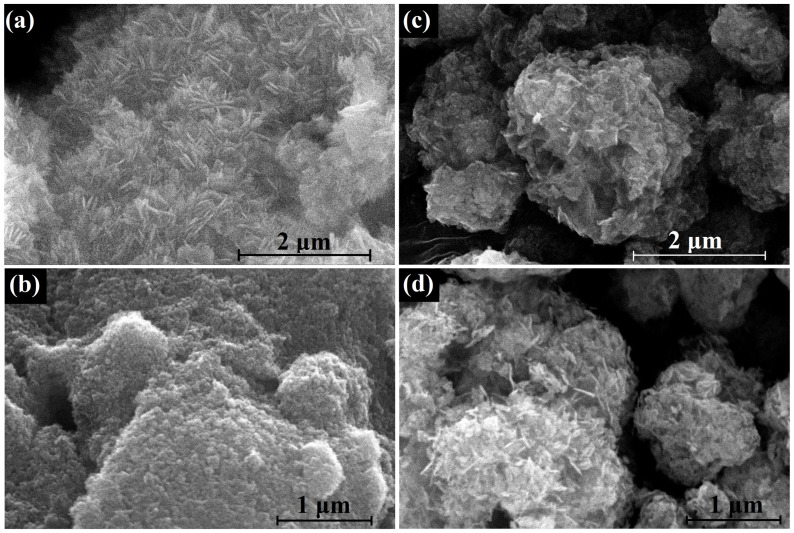
SEM images of synthesized HZ nanoplates (**a**) and nanocrystalline ZnO (**b**), obtained by their decomposition at 300 °C; commercial HZ powder (**c**) and ZnO (**d**), obtained by its decomposition at 300 °C.

**Figure 12 micromachines-14-00912-f012:**
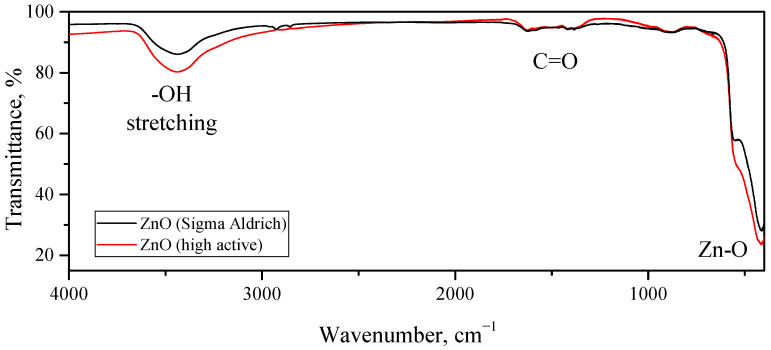
FTIR spectra of synthesized highly active ZnO (red line) and reference ZnO (black line) samples obtained by annealing of commercial (Sigma-Aldrich) hydrozincite powder.

**Table 1 micromachines-14-00912-t001:** BE positions and relative content of atoms in different charge states in the synthesized nanocrystalline ZnO.

BE Position, eV				Content, at.%			
$\mathrm{Zn}2p_{3/2}$	**O1s**	**C1s**	**N1s**	**Zn2p**	**O1s**	**C1s**	**N1s**
1021.3(I)	530.0(I)	284.8(I)	399.9	34.5(I)	23.7(I)	13.1(I)	0.1
1022.1(II)	530.8(II)	286.2(II)		3.9(II)	8.8(II)	2.6(II)	
	531.8(III)	288.8(III)			10.4(III)	2.9(III)	
				**38.4**	**42.9**	**18.6**	**0.1**

## Data Availability

The data presented in this study are available upon request from the corresponding author. The data are not publicly available due to privacy reasons.

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
