# Peer review of "Highly Active Nanocrystalline ZnO and Its Photo-Oxidative Properties towards Acetone Vapor"

_micromachines, 2023, doi:10.3390/mi14050912_

Round 1

Reviewer 1 Report

Reviewing your manuscript titled:  Highly active nanocrystalline ZnO and its photo-oxidative properties towards acetone vapor, I find it interesting, however, some points should be clarified and others improved before making a final decision.

- The summary should be improved, in its present form it is very qualitative, I suggest that the authors provide the main results obtained from the characterization techniques.

- The introduction of relevant background and research progress was not comprehensive enough.

- Why was this step necessary if the authors already had commercial ZnO? What are the authors looking for with adding distilled water to commercial zinc oxide? 50 g of commercial ZnO (ACS Reagent, ≥99.0%) was placed in a 3-L beaker and 1.75 L of deionized water was added to it. Resulting suspension was left stirred at 400 rpm using overhead stirrer at a temperature of 22–25â—¦ C for 30 min.

- The authors assure that they calculated the crystallite size, I didn't find the values nor comments about them.

- Put Miller indices in parentheses

- About XPS it is important to include some details about the settings. For example: What program was used to deconvolute the XPS spectra? The authors use a Lorentzian-Gaussian function for the fit? What is the G/L ratio used?

- Deconvolution of PL spectra is required. Provide details about the defects and their distribution in the samples.

Author Response

Reviewing your manuscript titled:  Highly active nanocrystalline ZnO and its photo-oxidative properties towards acetone vapor, I find it interesting, however, some points should be clarified and others improved before making a final decision.

Dear reviewer, I thank you for taking the time to review my manuscript, and I am glad that you appreciate it positively. I have done my best to answer your questions and make the necessary changes.

1. The summary should be improved, in its present form it is very qualitative, I suggest that the authors provide the main results obtained from the characterization techniques.

Answer: Thank you for your comment. Some results obtained from the characterization techniques were added in the abstract:

"The synthesized highly active ZnO powder has a mesoporous structure with a BET surface area of 79.5±4 m2/g, an average pore size of 20±2 nm and the cumulative pore volume is 0.507±0.051 cm3/g. The defect-related PL of the synthesized ZnO is represented by a broad band with a maximum at 575 nm."

(lines 9-12)

2. The introduction of relevant background and research progress was not comprehensive enough.

Answer: Thank you for your comment. We have expanded the Introduction section and added 19 new refs. The following text has been added:

"Acetone (2-propanone) is the simplest ketone, under normal conditions a highly volatile and flammable liquid. Acetone vapors are dangerous, firstly, for human health, and secondly, they are explosive when mixed with air in certain proportions. Typical symptoms of human exposure to acetone vapor are depression of the central nervous system and irritation of the mucous membranes of the eyes, nose and throat, which appear at concentrations starting from 237 ppm (574 mg/m3) when exposed for several hours. At the same time, the range of explosive concentration of acetone vapor in the air is 2.5--12.8 vol.%. The presence of acetone vapor in the air exhaled breath at a level of several ppm is a marker of various diseases, such as diabetes, so the detection of acetone at a low level is of great importance for non-invasive and quick medical diagnosis."

"Resistive type gas sensors are promising for detecting acetone in air, since they have high sensitivity, simple design, and a wide measuring range. One of the advanced gas sensitive materials for acetone detection is ZnO and materials based on it. At the same time, in the last decade, an approach using UV or visible irradiation instead of thermal activation to promote the gas sensitive properties of metal oxides has been widely developed."

"An analysis of the literature shows that studies of the gas-phase photooxidation of acetone in mainly focused TiO2 and TiO2-based materials, while information on the photooxidation of acetone on the ZnO surface is very limited."

"In the cited work, two samples of ZnO with an average crystallite size of 20.9--41.1 nm (BET surface area 49.8 m2/g) and 27.5--156.0 nm (BET surface area 13.2 m2/g) were synthesized. The study of the kinetics of photooxidation of acetone by gas chromatography was carried out according to the rate of release of the main oxidation product (CO2)."

Please see lines 54-87 for details.

3. Why was this step necessary if the authors already had commercial ZnO? What are the authors looking for with adding distilled water to commercial zinc oxide? 50 g of commercial ZnO (ACS Reagent, ≥99.0%) was placed in a 3-L beaker and 1.75 L of deionized water was added to it. Resulting suspension was left stirred at 400 rpm using overhead stirrer at a temperature of 22–25â—¦ C for 30 min.

Answer: Thank you for your comment. Commercial ZnO is micropowder, while we synthesized ZnO nanopowder. However, commercial ZnO is a good precursor for the synthesis of high-purity nano-ZnO, since it does not contain other elements than zinc and oxygen. Commercial ZnO was previously "soaked" at stirring in water to increase the hydration of the surface of the particles and enhance their reactivity. The following text has been added:

"to increase the hydration of the surface of the ZnO particles and enhance their reactivity" (lines 104-105)

4. The authors assure that they calculated the crystallite size, I didn't find the values nor comments about them.

Answer: Thank you for your comment. The crystallite size of the synthesized ZnO was calculated in two ways: 1) by the broadening of X-ray reflections (Scherrer equation, equation 3); 2) by measuring the crystallites on the TEM image (figure 6a), based on the results of which a particle size distribution diagram was upbuilded (figure 6c). The results obtained are discussed in section 3.2 (lines 271-283).

5. Put Miller indices in parentheses.

Answer: Thank you for your comment. We have placed the Miller indices in parentheses in Figure 6b. We checked all the text and did not find other places where this should be done.

6. About XPS it is important to include some details about the settings. For example: What program was used to deconvolute the XPS spectra? The authors use a Lorentzian-Gaussian function for the fit? What is the G/L ratio used?

Answer: Thank you for your comment. The following information has been added:

“The FitXPS soft was used for deconvolution of X-ray photoelectron spectra. The peaks were fiitted by Gaussian-Lorentzian functions with a variable ratio (G/L), and the background was determined by the Shirley function.” (lines 151-154)"

7. Deconvolution of PL spectra is required. Provide details about the defects and their distribution in the samples. 

Answer:  Thank you for your comment. The following text has been added:

“The observed PL peak can be resolved into at least two peaks with maxima at 521 and 593 nm, the intensity ratio of which is approximately 1:4.8. The photoluminescence of ZnO in the visible spectrum is due to point defects in its crystal structure and on the surface. Despite the fact that the interpretation of the PL spectra of ZnO is usually difficult due to the possibility of the existence of various defects in its structure, the green band (521 nm) can be attributed to the recombination of photoexcited holes in the valence band with electrons captured by ionized oxygen vacancies. The orange band (593 nm) is more likely to be related to the process of electron recombination at interstitial oxygen atoms”

(lines 298-306)

Reviewer 2 Report

This work introduced ZnO’s high photooxidative. This manuscript made a comprehensive illustration of crystal structure, Raman spectra, morphology, atomic charge state, optical and photoluminescence properties of samples. This paper is well written and organized, may be accepted after minor revisions solving questions as follow.

1.     Temperature plays an important role in photoluminescence, would authors please record the temperature of photoluminescence experiment.

Author Response

This work introduced ZnO’s high photooxidative. This manuscript made a comprehensive illustration of crystal structure, Raman spectra, morphology, atomic charge state, optical and photoluminescence properties of samples. This paper is well written and organized, may be accepted after minor revisions solving questions as follow.

Dear Reviewer, I thank you for the time you have taken to read my manuscript and I also appreciate your positive evaluation.

  1. Temperature plays an important role in photoluminescence, would authors please record the temperature of photoluminescence experiment.

Answer: Thank you for your comment. The following text has been added:

“PL spectra of samples were recorded at room temperature.” (line 163)

Reviewer 3 Report

In this article, Artem Chizhov et al. investigated the effect of nanocrystalline ZnO and it photooxidative characteristics on acetone vapors using mass spectroscopy and UV irritation. The idea and concept theme are quite interesting, and the manuscript is also well organized/written, which will likely attract attention and impact practical sensing applications. As such, I believe that the manuscript is suitable for publication; in which following minor amendments are needed before this work can be finally accepted for publication:

  • Provide FTIR and explain their chemical functional groups, since the along with the crystallinity of the as-synthesized ZnO and commercial ZnO there might be a small difference in the functional groups like OH for instance, which could help to make bonding at the time of acetone sensing.
  • Please provide the total optimized thickness of ZnO tablets.
  • What is the power density of UV light.
  • The Acetone vapor sensing mechanism needs to be emphasized in more detail.
  • Please explain the significance of ZnO as compared to other metals oxides, and other materials in the revised manuscript.
  • Please explain the importance of acetone sensing and its consequences. The concentration of acetone in this application of the sensor is unclear.

·         Please add some more references about recent sensors-based works to introduce in the introduction as below.

https://www.sciencedirect.com/science/article/pii/S0925838819317724#fig3

https://www.sciencedirect.com/science/article/pii/S092540052201783X

https://www.mdpi.com/1424-8220/16/11/1876

https://link.springer.com/article/10.1007/s12274-023-5472-x

  • How about long-term stability?
  • What is the role of UV-generated holes in sensing reactions?

Author Response

In this article, Artem Chizhov et al. investigated the effect of nanocrystalline ZnO and it photooxidative characteristics on acetone vapors using mass spectroscopy and UV irritation. The idea and concept theme are quite interesting, and the manuscript is also well organized/written, which will likely attract attention and impact practical sensing applications. As such, I believe that the manuscript is suitable for publication; in which following minor amendments are needed before this work can be finally accepted for publication:

Dear Reviewer, Thank you for taking the time to review my manuscript, and I am also pleased with your positive assessment of my manuscript. I tried to answer your questions exhaustively and make the necessary changes.

1. Provide FTIR and explain their chemical functional groups, since the along with the crystallinity of the as-synthesized ZnO and commercial ZnO there might be a small difference in the functional groups like OH for instance, which could help to make bonding at the time of acetone sensing.

Answer: Thank you for your comment. We have added the FTIR spectra of the samples discussed in Section 3.6 and commented on the resulting spectra. At the same time, it did not turn out that there is a dramatic difference between the content of OH-groups in the samples and, in our opinion, in this case it does not play a significant role in the mechanism of photooxidation. The following text has been added to the manuscript:

“Studies carried out by IR spectroscopy also demonstrate that the surface composition of the highly active synthesized ZnO and the reference ZnO are close to each other. The FTIR spectra of both samples are shown in the Figure 12. The main absorption band of the considered samples is located at 415 cm-1 (as well as a shoulder at 542  cm-1), which can be attributed to Zn-O stretching vibrations in the wurtzite crystal structure. Two weak absorption peaks at 1385 and 1625  cm-1belong to the symmetric and asymmetric vibrational modes of the carbonyl group, and the intensities of these peaks are very closed in both samples. The broad peak with a maximum at 3435  cm-1 refers to stretching vibrations of OH groups on the surface of ZnO grains, while the intensity of this band for highly active ZnO is somewhat higher than for the reference ZnO sample. This corresponds to a higher concentration of OH-groups on the surface of synthesized ZnO, which is consistent with its larger specific surface area compared to the reference ZnO sample.” (lines 451-464)

2. Please provide the total optimized thickness of ZnO tablets.

Answer: Thank you for your comment. The following information has been added:

"and 1 mm thick” (line 175)

3. What is the power density of UV light.

Answer: Thank you for your comment. Irradiance of samples was 40-60 mW/cm2. (lines 185-187)

4. The Acetone vapor sensing mechanism needs to be emphasized in more detail.

Answer:  Thank you for your comment. Indeed, this was a significant gap in our work. We have added the following text to the discussion:

“In a previous work, we developed a model according to which, under UV irradiation, the surface of ZnO is enriched with oxygen vacancies due to photolysis of the surface layers. The generated oxygen vacancies serve as adsorption sites for oxygen molecules, which then undergo dissociation with the healing of surface vacancies. Thus, the mechanism of acetone photooxidation on the surface of ZnO can be understood on the basis of the initial interaction of adsorbed acetone molecules with photoexcited holes acting as oxidizers, with the participation of oxygen atoms:

CH3COCH3 + 2h+ + 7O -> 3CO2 + 2H2O + 2H+

At the same time, to maintain electrical neutrality, protons are reduced by photoexcited electrons:

2H+ + 2e- ->H2

Although the formation of some intermediate or by-products of oxidation, such as formic acid, acetic acid, methylglyoxal, methanol, etc., could be expected as a result of the photooxidation of acetone, only the complete oxidation product was detected by mass spectrometry in the gas phase, indicating a strong oxidation potential photoexcited holes. The high efficiency of acetone photooxidation on the surface of the synthesized ZnO  is apparently due to the fact that, with a grain size of 10-16 nm, the surface exists under highly nonequilibrium conditions, which facilitates the formation of oxygen vacancies on the surface under UV irradiation.

(lines 400-418)

5. Please explain the significance of ZnO as compared to other metals oxides, and other materials in the revised manuscript.

Answer:  Thank you for your comment. The following information has been added:

"Compared to other metal oxides used as materials for gas sensors (SnO2, In2O3), ZnO has evidence photocatalytic properties, which, for example, was demonstrated using photoactivated isotopic oxygen exchange" (lines 27-29)

6. Please explain the importance of acetone sensing and its consequences. The concentration of acetone in this application of the sensor is unclear.

Answer:  Thank you for your comment. The following text has been added in the Introduction section:

"Acetone (2-propanone) is the simplest ketone, under normal conditions a highly volatile and flammable liquid. Acetone vapors are dangerous, firstly, for human health, and secondly, they are explosive when mixed with air in certain proportions. Typical symptoms of human exposure to acetone vapor are depression of the central nervous system and irritation of the mucous membranes of the eyes, nose and throat, which appear at concentrations starting from 237 ppm (574 mg/m3) when exposed for several hours. At the same time, the range of explosive concentration of acetone vapor in the air is 2.5--12.8 vol.%. The presence of acetone vapor in the air exhaled breath at a level of several ppm is a marker of various diseases, such as diabetes, so the detection of acetone at a low level is of great importance for non-invasive and quick medical diagnosis." (lines 55-64)

7. Please add some more references about recent sensors-based works to introduce in the introduction as below.

Answer:  Thank you for your comment. These and some other references (total 19 new references) have been added to the Introduction section.

8. How about long-term stability?

Answer:  Thank you for your comment. Since our work is devoted, rather, to the study of the mechanisms of processes, the study of long-term stability has not yet been carried out.

9. What is the role of UV-generated holes in sensing reactions?

Answer:  Thank you for your comment. The role of holes is shown in the mechanism of acetone photooxidation proposed by us (lines 400-418); holes act as oxidizers in the process of acetone photooxidation.

Round 2

Reviewer 1 Report

This version of the manuscript can be accepted for publication.